# Association of TLR9-1237T>C; rs5743836 polymorphism with increased risk of Hodgkin's lymphoma: A case-control study

Sohaib Al-Khatib[1☯]*, Amin Shabaneh[2☯], Nour Abdo[3‡], Laith AL-Eitan[4‡], Abdel-Hameed Al-Mistarehi[5‡], Yousef Khader[6‡]

1 Faculty of Medicine, Department of Pathology and Laboratory Medicine, Jordan University of Science and Technology, Irbid, Jordan, 2 Faculty of Medicine, Jordan University of Science and Technology, Irbid, Jordan, 3 Faculty of Medicine, Department of Public Health, Jordan University of Science and Technology, Irbid, Jordan, 4 Faculty of Science and Arts, Department of Biotechnology and Genetic Engineering, Jordan University of Science and Technology, Irbid, Jordan, 5 Faculty of Medicine, Department of Family Medicine, Jordan University of Science and Technology, Irbid, Jordan, 6 Faculty of Medicine, Department of Public Health and Community Medicine, Jordan University of Science and Technology, Irbid, Jordan

☯ These authors contributed equally to this work.
‡ NA, LAE, AHAM and YK also contributed equally to this work.
* smkhatib4@just.edu.jo

**Data Availability Statement:** All relevant data are within the paper and its Supporting Information files.

## Abstract

Mature B-cell neoplasms are typically divided into Hodgkin and Non-Hodgkin Lymphomas. Hodgkin Lymphoma is characterized by the neoplastic Reed-Sternberg cells, usually harbored in an inflammatory background, with a frequent clinical presentation of mediastinal lymphadenopathy. Many studies link between autoimmunity and lymphomagenesis, a large proportion of these studies evidently trace the pathogenesis back to the misdirected detection of self-derived nucleic acids by Toll-Like Receptors (TLRs), especially those of the intracellular type. In this study, we analyzed the relationship between a selected SNP in TLR9 (TLR9-1237T>C; rs5743836) and the risk and overall survival of HL patients in a Jordanian Arab population. A total of 374 subjects; 136 cases of Hodgkin lymphoma and 238 matched healthy controls were incorporated in this study. Genomic DNA was extracted from formalin-fixed paraffin-embedded tissues. Genotyping of the genetic polymorphisms was conducted using a sequencing protocol. The results show a statistically significant higher distribution of the rs5743836 (TLR9-1237T>C) allele among the case population, with a p-value of 0.031 (<0.05). This distribution proved significant when studied in the codominant (only significant in the T/C genotype, p-value = 0.030), dominant (p-value = 0.025), and overdominant (p-value = 0.035) models. None of the models showed any statistically significant difference in survival associated with the rs5743836 (TLR9-1237T>C) SNP.

## Introduction

Mature B-cell neoplasms are typically divided into Hodgkin and Non-Hodgkin Lymphomas (HL and NHL) [1]. Unlike NHL which added estimated 2.78% to the global cancer burden

**Funding:** The study was supported by the Jordan University of Science and Technology Grant number 20170225 The funders had no role in study design, data collection and analysis, decision to publish, or preparation of the manuscript.

**Competing interests:** The authors have declared that no competing interests exist.

(510,000 new cases in the year 2018), HL is considered an uncommon B-cell malignancy accounting for 0.4% of all new tumors and 0.3% of all new cancer deaths in the year 2018 [2, 3]. The incidence of HL shows variable trends that are significantly influenced by geographic, socioeconomic, racial, gender, and age differences [4]. Incidence is higher in more developed countries and among males, characteristically displaying a bimodal curve of incidence that is most significant for young adults (of age 15–34) and those above 50 years of age [5, 6]. Epidemiologic studies indicate that genetic and environmental factors both play essential roles in the pathogenesis of HL [7]. Among the most highlighted environmental factors is Epstein-Barr viral infection, which is clearly evident in the countless numbers of studies outlining its prominent but poorly elucidated role in the pathogenesis of a large proportion of HL cases [8–11].

HL is characterized by the neoplastic Reed-Sternberg cells, usually harbored in an inflammatory background, with a frequent clinical presentation of mediastinal lymphadenopathy. It is morphologically and immunohistochemically subcategorized into classical Hodgkin lymphoma (CHL) and Nodular lymphocyte predominant Hodgkin lymphoma (NLPHL). The majority of HL (around 90%) are of the classical type, while the rest are of the more indolent and slow-progressing nodular lymphocyte predominant type. CHL is further divided into nodular sclerosis (NSCHL–the most common subtype), mixed cellularity (MCCHL–the second most common subtype), lymphocyte rich (LRCHL–the subtype with the best prognosis), and lymphocyte depleted (LDCH–the rarest subtype with the most aggressive prognosis). LDCHL and MCCHL have prognoses that are significantly worse than that of NSCHL, being commonly associated with immunosuppression such as that seen in HIV infection. MCCHL subtype shows the most affiliation with EBV infection; approximately 75% of this subtype's cases show evidence of EBV-expressed nuclear RNA transcripts and latent membrane protein 1 (LMP1) [12].

Many studies link between immune dysregulation and resultant risk of lymphomagenesis [13–15]. Other studies outline links between autoimmunity and lymphomagenesis, a large proportion of these studies evidently trace the pathogenesis back to the misdirected detection of self-derived nucleic acids by Toll-Like Receptors (TLRs), especially those of the intracellular type [16–18]. As in almost all other cancers, the inflammatory milieu set forward by the immune response is important in the pathogenesis and progression of HL. This importance has been noted in the evident interactions between Reed-Sternberg cells and the background inflammatory cell environment [19]. A huge variety of growth factors, chemokines and cytokines exists in this environment and play an essential role in the progression of the disease [20]. The local immune reaction in HL has been previously addressed as the most prominent among all tumors, with a few exceptions [21]. It is important to note however that CHL resembles a spectrum, and that LDCHL is an aggressive subtype partly due to the absence of an adequate immune microenvironment, which means an absence of an effective anti-tumor response [21].

A particular single nucleotide polymorphism (SNP) on Toll-Like Receptor 9 (TLR9) has been the center of multiple studies that analyzed its relation to lymphoma formation; this SNP is rs5743836 (TLR9-1237T>C). In the aforementioned studies, results have demonstrated that this SNP either shows no association with the risk of HL development or significantly increased the risk of developing HL and NHL [22–24].

Tumor development, growth and response to treatment is influenced by the surrounding proinflammatory microenvironment. TLR activation plays an important role in immune-response mediation [25, 26]. Disruption of TLR function promotes cancer development by evading immune system, and over activation of TLR may play an important role in inhibition of cancer evolution [27]. CPGs motifs are commonly found in bacterial DNA but not

vertebrate DNA. Additionally, CpGs motifs in bacterial DNA are mostly un-methylated. The recognition of un-methylated foreign CPGs motifs in bacterial DNA and triggering an innate immune response occurs through TLR9 protein [28, 29]. Normal B lymphocytes and neoplastic lymphoma cells of small lymphocytic lymphoma, mantle cell lymphoma, follicular lymphoma and diffuse large be cell lymphoma, all have been reported to express TLR9 [30, 31]. CpG oligonucleotides are considered TLR9 agonist and stimulate TLR9 expression. Signaling through TLR9 stimulation by exogenous or endogenous ligands plays a fundamental role in host immune-response and subsequently lymphoma risk [32]. Signaling of TLR pathways are basically NF-kB dependent [33]. People carrying the variant "C" allele of rs5743836 TLR9 promoter polymorphism (TLR9-123C) show increased TLR9 mRNA expression, transcriptional activity and deregulated immunological response secondary to increased binding affinity of TLR9 to NF-κB by creation of a potential NF-κB binding site [34].

The aim of this study was to examine the relationship between a selected SNP in TLR9 (TLR9-1237T>C) and the risk and overall survival of HL patients in a Jordanian Arab population.

## Material and methods

### Patients and data collection

This is a retrospective cross-sectional study examine the relationship between rs5743836 (TLR9-1237T>C) polymorphism and the risk and survival of Hodgkin lymphoma patients among an Arab Jordanian population. The study population was composed of one hundred and thirty-six (136) patients and two hundred and thirty-eight (238) healthy cancer-free control subjects with similar geographic and ethnic backgrounds to the patients. The 136 cases of HL were retrieved from the archives of King Abdullah University Hospital during the period of 2013 to 2019. All cases were reviewed by (SK) and one representative section was chosen from each case. The human ethics approval was attained by the ethical committee of Jordan University of Science and Technology [Institutional Review Board (IRB) code number 5/106/2017, dated 8/06/2017] in accordance with the 1964 Declaration of Helsinki and its later amendments. Formal written informed consent was not required with a waiver by the IRB. All control subjects were voluntarily involved and signed written informed consent. Cases' and controls' names were coded and blinded and treated confidentially. For the patients (minors and adults); formal written informed consent was not required with a waiver by the IRB. All control subjects were voluntarily involved and a signed written informed consent was obtained by all control subjects or their parents/guardians (for minors). Cases' and controls' names were coded and blinded and treated confidentially.

### DNA analysis

The commercially available DNeasy Blood & Tissue Kit (Qiagen Ltd., West Sussex, UK) was used, and according to manufacturer's protocol, to DNA extraction for the HL patients from formalin-fixed paraffin-embedded tissue. Genomic DNA from control-subjects' blood samples was extracted using the QIAamp® or Promega DNA Mini Kit according to the manufacturer's instruction. The quality of extracted DNA was examined using agarose gel electrophoresis and ethidium bromide staining. The concentration and purity of extracted DNA were assessed using a NanoDrop 1000® spectrophotometer. The SNP was genotyped using the Sequenom MassARRAY® system (iPLEX GOLD). The pure DNA samples with their concentrations were sent to the Australian Genome Research Facility (AGRF, Melbourne Node, Melbourne, Australia) for genotyping of SNP rs5743836 (TLR9-1237T>C) in all subjects (patients and controls). The SNP, SNP's position, and primer sequences for TLR9 gene are shown in

**Table 1. The SNP, SNP's position and primer sequences for TLR9 gene.**

| SNP-ID | Gene | Chr^ | bp* | Primer Forward | Primer Reverse |
|--------|------|------|-----|----------------|----------------|
| rs5743836 | TLR9 | 3 | 52226766 | ACGTTGGATGTTGGGATGTGCTGTTCCCTC | ACGTTGGATGAGCAGAGACATAATGGAGGC |

* bp: base pair (Genomic Position).

^ Chr: Chromosome.

Table 1. Genotyping with the Sequenom MassARRAY® system (iPLEX GOLD) (Sequenom, San Diego, CA, USA) was performed at the AGRF according to the manufacturer's recommendations (Sequenom, San Diego, CA, USA). Genotype distributions were compared between patients and controls. Unconditional logistic regression analysis was used to estimate the association between the genotype frequency and the risk of developing HL.

## Statistical analysis

Overall survival (OS) was calculated from the date of diagnosis to the date of death or the last visit for those who were alive at the time of final data collection and analysis. The Statistical Package for Social Sciences IBM SPSS Statistics for Windows version 20.0 (IBM Corp., Armonk, NY, USA) was used to identify the genotypic and allelic associations. The clinical characteristics and response rate of the patients were compared using chi-square tests. The Hardy–Weinberg equilibrium (HWE) test was estimated by a goodness-of-fit $\chi^2$ test. The Kaplan–Meier method was used to construct survival curves, and the results were compared using a log-rank/Wilcoxon (Gehan) statistic. The association between polymorphism and the risk for HL was calculated using unconditional logistic regression. The survival curves were displayed using Graph Pad Prism 6 software. All significant variables ($p < 0.05$) were entered into a multivariate model to adjust for possible confounders.

## Results

### Demographic and clinical characteristics of the study population

A total of 374 subjects were included in the study and their DNA samples collected. Of these 374 subjects, 136 were cases (HL patients) and 238 were controls. The case group had a mean age (and range) of 30.7 years (3–78), and was comprised 87 males (64%) males and 49 females (36%). On the other hand, the control group had mean age (and range) of 43.2 years (6–89), and was contained 92 males (38.7%) and 146 females (61.3%). The case group had a lower age mean and a higher proportion of males. Of the 374 DNA samples, 10 (6 from case group and 4 from control group) didn't yield sequencing results and were omitted from the statistical tests. Eventually 130 cases and 234 controls were statistically tested. The 130 HL cases were of the following histological subtypes: mixed cellularity (MC– 58 cases [45%]), nodular sclerosis (NS– 52 cases [40%]), lymphocyte rich (LR– 14 cases [11%]), lymphocyte depleted (LD– 3 [2%]), and finally the non-classical nodular lymphocyte predominant (NLPHL– 3 [2%]). The demographic and clinical data of all subjects is summarized in Table 2.

### Genotype distributions and association with risk of HL

The distribution of rs5743836 (TLR9-1237T>C) in both the HL patients and the control group was in Hardy-Weinberg Equilibrium (HWE), with a p-value > 0.05 (p = 0.34 for control group). This eliminates the chance of stratification error during genotyping.

The results show how a statistically significant higher frequency of rs5743836 allele T among patients, with a p-value of 0.02 (<0.05). This distribution proved significant when

**Table 2. Demographic and clinical data of the 136 cases and 238 controls enrolled in the study.**

| Demographic Data | Cases N (%) | Controls N (%) |
|---|---|---|
| **Gender** | | |
| Male | 87 (64) | 92 (38.7) |
| Female | 49 (36) | 146 (61.3) |
| **Age in Years*** | | |
| 0–14 | 23 (16.9) | 3 (1.3) |
| 15–19 | 19 (14) | 18 (7.6) |
| 20–40 | 54 (39.7) | 89 (37.4) |
| 41–55 | 26 (19.1)) | 59 (24.8) |
| >55 | 14 (10.3) | 69 (28.9) |
| | | |
| Mean (Range) | 30.735 (3–78) | 43.2 (6–89) |
| Median (IQR) | 28 (18–43) | 44 (24.2–57) |
| **Clinical Data** | | |
| **Survival Status** | | |
| Alive | 118 (86.8) | ---- |
| Dead | 18 (13.2) | ---- |
| **Survival (Months)** | | ---- |
| Median | Not reached** | ---- |
| **Type of Lymphoma** | | ---- |
| LR | 15 (11) | ---- |
| MC | 61 (44.9) | ---- |
| NS | 55 (40.4) | ---- |
| LD | 2 (1.5) | ---- |
| NLPHL | 4 (2.2) | ---- |
| **B-Symptoms** | | |
| Yes | 23 (16.9) | ---- |
| No | 67 (49.3) | ---- |
| No data found | 5 (3.7) | |
| Missing | 41 (30.1) | |
| **Serum Albumin (g/L)** | | ---- |
| Mean (Range) | 38.111 (15–54.2) | ---- |
| Median (IQR) | 39 (34–42.85) | ---- |
| **Serum LDH (IU/L)** | | |
| Mean (Range) | 546.559 (207–1561) | ---- |
| Median (IQR) | 486 (374–677) | ---- |
| **ESR (mm/hr)** | | |
| Mean (Range) | 63.714 (2–153) | ---- |
| Median (IQR) | 61.5 (30.5–100) | ---- |

LR: Lymphocyte rich; MC: Mixed cellularity; NS: Nodular Sclerosis; LD: Lymphocyte Depleted; NLPHL: Nodular lymphocyte-predominant Hodgkin's lymphoma; ESR: Erythrocyte sedimentation rate.

*Age of controls vs age at diagnosis in cases

**More than half of the cases population was alive at time of analysis.

studied in the codominant (only significant in the A/G genotype, p-value = 0.030), dominant (p-value = 0.025), and overdominant (p-value = 0.035) models. The odds ratio (OR) value was **1.78** (95% confidence interval [1.08, 2.93]), **1.77** (95% confidence interval [1.10, 2.86]), and

**Table 3. Allele and genotype frequency in cases and controls.**

| SNP rs5743836 | Case: N (%) | Control: N (%) | P-value |
|---|---|---|---|
| Allele T | 213 (82) | 412 (88) | 0.031 |
| Allele C | 47 (18) | 56 (12) | |
| Genotype T/T | 87 (66.9) | 183 (78.2) | 0.062 |
| Genotype T/C | 39 (30) | 46 (19.7) | |
| Genotype C/C | 4 (3.1) | 5 (2.1) | |

**1.75** (95% confidence interval [1.07, 2.87]), respectively. These results support an almost two-fold increase in risk of developing HL associated with the rs5743836 (TLR9-1237T>C) SNP. Findings are summarized in Tables 3 and 4.

## Genotype distributions and association with survival

None of the models showed any statistically significant difference in survival associated with the rs5743836 (TLR9-1237T>C) SNP (Fig 1).

## Discussion

This study investigated a TLR9 SNP and its association with both the risk and prognosis of HL. When the SNP rs5743836 (TLR9-1237T>C) was examined among both the case and the control groups, our results showed an almost two-fold risk of HL associated with this genotype, statistically significant in the dominant and the overdominant models. However, statistical analysis showed no significant impact of this SNP on the prognosis of HL in patients enrolled in the study.

Due to TLRs' established role in both the immune response and the progression of cancers, as well as TLR9's established role in some autoimmune inflammatory diseases, many studies have investigated the role of genetic polymorphisms in TLR9 and their corresponding effects on risk of cancer development [22, 23, 35]. Some of these studies have investigated this particular variable specifically in the context of the risk of developing HL and NHL [22, 23]. Fewer studies have investigated the associated risk of TLR9 polymorphism on lymphoma development in the Arab region [36]. However, there seems to be no studies examining the impact of TLR9 polymorphisms on both the risk and prognosis of HL patients, especially among an Arab population.

SNP rs5743836 (TLR9-1237T>C) was the center of many studies investigating its association with the risk of developing both HL and NHL [22, 23]. A study demonstrated a

**Table 4. Testing the association between risk of HL and SNP rs5743836 by finding odds ratio (OR) for different genetic models with confidence interval (CI) of 95%.**

| Genetic model | Genotype | Case: N (%) | Control N (%) | OR (95% CI) | P-value |
|---|---|---|---|---|---|
| Codominant | T/T | 87 (66.9) | 183 (78.2) | 1 | |
| | T/C | 39 (30) | 46 (19.7) | 1.78 (1.08–2.93) | 0.03 |
| | C/C | 4 (3.1) | 5 (2.1) | 1.68 (0.44–6.42) | 0.683 |
| Dominant | T/T | 87 (66.9) | 183 (78.2) | 1 | |
| | T/C-C/C | 43 (33.1) | 51 (21.8) | 1.77 (1.10–2.86) | 0.025 |
| Recessive | T/T-T/C | 126 (96.9) | 229 (97.9) | 1 | |
| | C/C | 4 (3.1) | 5 (2.1) | 1.45 (0.38–5.51) | 0.84 |
| Overdominant | T/T-C/C | 91 (70) | 188 (80.3) | 1 | |
| | T/C | 39 (30) | 46 (19.7) | 1.75 (1.07–2.87) | 0.035 |

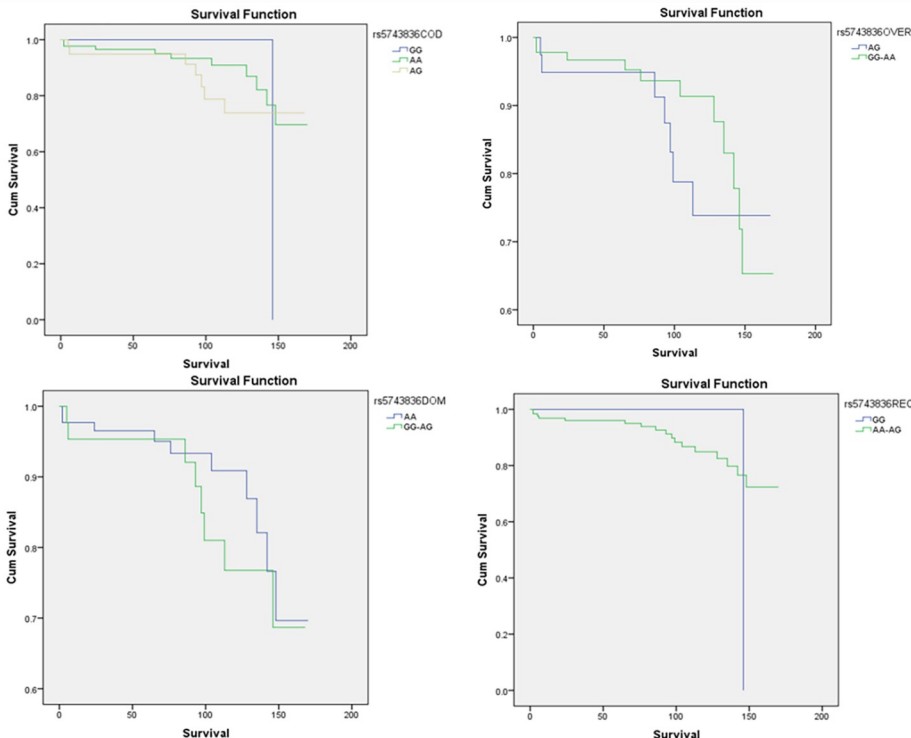

**Fig 1. Overall survival (OS) for 130 HL patients according to SNP rs5743836 in different genetic models.** Top row from left to right: (a) Comparison among CC, TT, and CC in the codominant model (p-value = 0.58). (b) Comparison among TC and CC/TT in the overdominant model (p-value = 0.42). Bottom row from left to right: (a) Comparison among TT and CC/TC in the dominant model (p-value = 0.66). (b) Comparison among CC and TT/TC in the recessive model (p-value = 0.32). All p-values indicated were Wilcoxon (Gehan) Statistic.

population-dependent impact of this SNP on developing NHL, as the Italian and the Portuguese had a significant risk (odds ratio) of 1.84 and 1.85 respectively (p-values: 7.3E−9 and 7.3E−9 respectively), while no significant correlation was seen in the US cohort of NHL patients [22]. On the other hand, another study showed that this SNP had a significant odds ratio (p-value = 0.01) of 1.99 (95% confidence interval [1.18–3.36]) of developing HL in the Greek population [23]. Both of these studies support the clearly identifiable link between the TLR9 immunity component and lymphomagenesis. Although in the context of this paper, the study of HL among the Greek population provides more supportive value as it shows a two-fold risk associated with this SNP, as was found when statistically analyzing our results.

TLRs are a family of receptors that function in the innate immune response and are part of the earliest surveillance methods that respond to infection. Humans are known to possess a family that consists of 10 different TLRs [37]. These early immune receptors are also given the name pattern-recognition receptors (PRRs) due to their specialized role in detecting pathogen-associated molecular patterns (PAMPs) as well as damage-associated molecular patterns (DAMPs). PAMPs are displayed by infectious organisms while DAMPs are self-derived molecules released from damaged cells. TLRs are expressed by phagocytic innate immune cells such as macrophages and dendritic cells (DCs), as well as non-immune cells which include epithelial cells and fibroblasts. This family of receptors is further divided into two subfamilies: intracellular TLRs and cells surface TLRs. Intracellular TLRs (3, 7, 8 and 9 in humans) primarily act on viral detection through the recognition of foreign nucleic acids. Recent studies have found that these actions can be misdirected towards self-derived nucleic acids in autoimmune

diseases such as systemic lupus erythematosus (SLE) and psoriasis [17, 18]. TLR9 in particular is concerned with recognizing non-methylated CG dinucleotides (CpG DNA motifs), which are extremely more common in prokaryotic cells than in eukaryotic cells, hence this receptor chiefly controls responses to bacterial DNA as well as DNA of some viruses [38].

In general, TLRs' role has been described as a double edged sword in the development of cancers; DAMPs alert the immune system to the presence of neoplastic cells while simultaneously inducing chronic inflammation that can escalate the cancer's progression [39]. Furthermore, one of the main pathogenic features of HL is NF-kB activation, which is directly induced by EBV's LMP1 and indirectly stimulated by TLR9's response to EBV infection [23, 40]. This may provide solid ground for justifying the significant association of TLR9 with development of HL.

## Conclusion

In this study, we demonstrated in our Jordanian Arab sample that the risk of HL was almost two-fold in those with the SNP rs5743836 (TLR9-1237T>C), when examined in the dominant, codominant and overdominant models (p-values are 0.025, 0.030 and 0.035 respectively). The study found no significant association between this SNP and the overall survival of HL patients in the sample studied.

In spite of the significant associations between the SNP rs5743836 (TLR9-1237T>C) and the development HL that were discovered in our study and further supported by other studies as discussed previously, several weaknesses existed in the methodology of this study. Firstly, the loss of data–that was discussed previously–from both the control and case groups in unequal amounts, either due to invalid DNA sampling results or loss of sample, has potential to weaken the findings. Secondly, the small sample size of 374 subjects (of which 136 were cases and 238 were controls), which can make it questionable to infer the predicted association on the general population. To put this in context: to confidently (CI = 95%) detect a relatively small influence (OR = 1.2) of a genetic polymorphism present in 10% of the population, 15 000 case-control pairs at least are to be included in a single study [41]. This large number of patients is not comparable to the number included in the study, and is largely beyond the capabilities of the medical institution this research study was conducted at (and many others for that matter). A possible improvement to this is conducting meta-analyses in the future, where pooled data from collaborative studies would provide tremendous numbers of subjects that would enhance the significance of inferred findings [41].

## Supporting information

**S1 Dataset.**
(XLSX)

## Author Contributions

**Conceptualization:** Sohaib Al-Khatib, Laith AL-Eitan.

**Data curation:** Sohaib Al-Khatib.

**Formal analysis:** Amin Shabaneh, Nour Abdo, Yousef Khader.

**Funding acquisition:** Sohaib Al-Khatib.

**Methodology:** Laith AL-Eitan.

**Resources:** Laith AL-Eitan, Abdel-Hameed Al-Mistarehi.

**Supervision:** Sohaib Al-Khatib.

**Writing – original draft:** Sohaib Al-Khatib, Amin Shabaneh.

**Writing – review & editing:** Sohaib Al-Khatib, Amin Shabaneh, Nour Abdo, Yousef Khader.

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
