## [Decision Letter · Decision Letter 0]

23 May 2022

PONE-D-22-13603The impact of SNP rs5743836 of TLR9 on the risk and prognosis of Hodgkin’s Lymphoma: a case-control study PLOS ONE

Dear Dr. Al Khatib,

Thank you for submitting your manuscript to PLOS ONE. After careful consideration, we feel that it has merit but does not fully meet PLOS ONE’s publication criteria as it currently stands. Therefore, we invite you to submit a revised version of the manuscript that addresses the points raised during the review process.

We look forward to receiving your revised manuscript.

Kind regards,

Alvaro Galli

Academic Editor

PLOS ONE

Journal Requirements:

 The study was supported by the Jordan University of Science and Technology Grant number 20170225   

Please include this amended Role of Funder statement in your cover letter; we will change the online submission form on your behalf

Reviewers' comments:

Reviewer's Responses to Questions

**Comments to the Author**

1. Is the manuscript technically sound, and do the data support the conclusions?

Reviewer #1: Yes

Reviewer #2: Yes

2. Has the statistical analysis been performed appropriately and rigorously? 

Reviewer #1: No

Reviewer #2: Yes

3. Have the authors made all data underlying the findings in their manuscript fully available?

Reviewer #1: Yes

Reviewer #2: Yes

4. Is the manuscript presented in an intelligible fashion and written in standard English?

Reviewer #1: Yes

Reviewer #2: Yes

5. Review Comments to the Author

Reviewer #1: In this manuscript Al-Khatib et al. evaluated the frequency of TLR9 SNP rs5743836 in Hodgkin Lymphoma patients. Although scientifically sounds, the manuscript has several points that should be addressed:

The title of the manuscript is misleading since no association of the evaluated SNP was observed concerning HL prognosis.

Although authors claim that “particular single nucleotide polymorphism (SNP) on Toll-Like Receptor 9 (TLR9) has been the center of multiple studies that analyzed its relation to lymphoma formation; this SNP is rs5743836 (TLR9-1237T>C). In the aforementioned studies, results have demonstrated that this SNP significantly increased the risk of developing HL and NHL.” they forgot to mention references that find no association if this specific SNP and HL (see, for example, Martin et al. Leuk Lymphoma. 2017 Feb;58(2):438-444. doi: 10.1080/10428194.2016.1190972.)

What should be the connection between rs5743836 and HL? Why to evaluate this and not other TLR9 variants (or any other gene variants)? The question is: What is the biological reason for choosing this gene variant? All these points were only superficially approached in the Discussion, but should be deeply presented in Introduction.

I was quite puzzled by the nomenclature used. Why authors refer to SNP TLR9-1237T>C but A/G genotypes? Please, standardize the manuscript nomenclature. It is quite difficult to follow the results comparing allele versus genotype (in fact, there is some confusion between alleles and genotypes all along the manuscript, please revise this point).

Please, revise the manuscript concerning the way results are described. For example, instead of saying “...a statistically significant higher distribution of the rs5743836”, I would suggest to say “how a statistically significant higher frequency of rs5743836 allele A among patients”.

In fact, a qui-square testing with Yate’s correction resulted in a p = 0.031. Please, review calculations.

Considering the acknowledged relation of EBV infection and HL development, I would expect data concerning EBV positivity in this cohort.

As highlighted by the authors, sample size is extremely low. As comparison, the reported study from Carvalho et al, back in the 2012’s included three distinct cohorts encompassing respectively 797 Portuguese, 494 Italian and 868 US patients. In this context, besides significance, I wonder about the originality/novelty of the present study.

Reviewer #2: Dear Author,

Please mention the possible relationship between genotypes and clinical parameters. And also you should give information about how this base change affects the TLR-9 gene function.

Best Regards

6. PLOS authors have the option to publish the peer review history of their article (what does this mean?). If published, this will include your full peer review and any attached files.

Reviewer #1: No

Reviewer #2: No

---

## [Author Response · Author response to Decision Letter 0]

4 Jul 2022

June 26, 2022

Academic editor, 

RE: Manuscript PONE-D-22-13603

Dear Alvaro Galli.

Academic Editor

PLOS ONE

Thank you for giving me the opportunity to submit a revised draft of my manuscript titled “The impact of SNP rs5743836 of TLR9 on the risk and prognosis of Hodgkin’s Lymphoma: a case-control study”. We greatly appreciate the reviewers for their complimentary, insightful comments and suggestions. We have carried out the changes that the reviewers suggested and revised the manuscript accordingly.

Please find the attached point-by-point response to reviewers concerns.

We hope that you find our responses satisfactory and that the manuscript is now acceptable for publication. 

Sincerely

Sohaib M. Al-Khatib, M.D.

Associate Professor

Department of Pathology and Laboratory Medicine

Jordan University of Science and Technology

E-mail: smkhatib4@just.edu.jo

Response to Journal Additional requirements

We appreciate the respected comments. The following are our point-by- point responses.

Point 1: Please ensure that your manuscript meets PLOS ONE's style requirements, including those for file naming.

Response 1: Efforts taken to make sure that the manuscript meets PLOS ONE’s style requirements. 

Point 2: You indicated that you had ethical approval for your study. In your Methods section, please ensure you have also stated whether you obtained consent from parents or guardians of the minors included in the study or whether the research ethics committee or IRB specifically waived the need for their consent.

Response 2: Formal written informed consent was not required with a waiver by the IRB. All control subjects were voluntarily involved and signed written informed consent. Cases’ and controls’ names were coded and blinded and treated confidentially. The Methods section included the above statement (I. 137-140 in revised version with track changes and l. 126-129 in corrected clean version).

Point 3: Thank you for stating the following financial disclosure: The study was supported by the Jordan University of Science and Technology Grant number 20170225. Please state what role the funders took in the study. If the funders had no role, please state: "The funders had no role in study design, data collection and analysis, decision to publish, or preparation of the manuscript." 

Response 3: "The funders had no role in study design, data collection and analysis, decision to publish, or preparation of the manuscript." 

Response to Reviewer 1 Comments

We appreciate the respected reviewer comments. The following are our point-by- point responses.

Point 1: The Title of the manuscript is misleading since no association of the evaluated SNP was observed concerning HL prognosis.

Response 1: Agree with the respected reviewer. I have, accordingly, modified the Title into “Association of TLR9-1237T>C; rs5743836 polymorphism with increased risk of Hodgkin’s Lymphoma: a case-control study” (I. 1-2 in revised version with track changes and l. 1-2 in clean corrected version).

Point 2: Although authors claim that “particular single nucleotide polymorphism (SNP) on Toll-Like Receptor 9 (TLR9) has been the centre of multiple studies that analysed its relation to lymphoma formation; this SNP is rs5743836 (TLR9-1237T>C). In the aforementioned studies, results have demonstrated that this SNP significantly increased the risk of developing HL and NHL.” they forgot to mention references that find no association of this specific SNP and HL (see, for example, Martin et al. Leuk Lymphoma. 2017 Feb;58(2):438-444. doi: 10.1080/10428194.2016.1190972.)

Response 2: Thank you for pointing this out. I agree with this comment. Therefore, the following sentence about studies and corresponding references that find no association of SNP rs5743836 (TLR9-1237T>C) and HL has been included in the Introduction (I. 99-100 in revised version with track changes and l. 87-89 in clean corrected version)

“In the aforementioned studies, results have demonstrated that this SNP either shows no association with the risk of HL development or significantly increased the risk of developing HL and NHL (22-24)”

Point 3: What should be the connection between rs5743836 and HL? Why to evaluate this and not other TLR9 variants (or any other gene variants)? The question is: What is the biological reason for choosing this gene variant? All these points were only superficially approached in the Discussion, but should be deeply presented in Introduction.

Response 3: Thank you for pointing this out. The following paragraph which answers the reviewer’s points has been included in the Introduction (I. 101-118 in revised version with track changes and l. 90-107 in clean corrected version).

“Tumor development, growth and response to treatment is influenced by the surrounding proinflammatory microenvironment. TLR activation plays an important role in immune-response mediation (25, 26). Disruption of TLR function promotes cancer development by evading immune system, and over activation of TLR may play an important role in inhibition of cancer evolution(27). CPGs motifs are commonly found in bacterial DNA but not vertebrate DNA. Additionally, CpGs motifs in bacterial DNA are mostly un-methylated. The recognition of un-methylated foreign CPGs motifs in bacterial DNA and triggering an innate immune response occurs through TLR9 protein (28, 29). Normal B lymphocytes and neoplastic lymphoma cells of small lymphocytic lymphoma, mantle cell lymphoma, follicular lymphoma and diffuse large be cell lymphoma, all have been reported to express TLR9 (30, 31). CpG oligonucleotides are considered TLR9 agonist and stimulate TLR9 expression. Signaling through TLR9 stimulation by exogenous or endogenous ligands plays a fundamental role in host immune-response and subsequently lymphoma risk (32). Signaling of TLR pathways are basically NF-kB dependent (33). People carrying the variant “C” allele of rs5743836 TLR9 promoter polymorphism (TLR9-123C) show increased TLR9 mRNA expression, transcriptional activity and deregulated immunological response secondary to increased binding affinity of TLR9 to NF-κB by creation of a potential NF-κB binding site (34)”.

Point 4: I was quite puzzled by the nomenclature used. Why authors refer to SNP TLR9-1237T>C but A/G genotypes? Please, standardize the manuscript nomenclature. It is quite difficult to follow the results comparing allele versus genotype (in fact, there is some confusion between alleles and genotypes all along the manuscript, please revise this point).

Response 4: Agree with the respected reviewer. The manuscript nomenclature has been standardized.

Point 5: Please, revise the manuscript concerning the way results are described. For example, instead of saying “...a statistically significant higher distribution of the rs5743836”, I would suggest to say “how a statistically significant higher frequency of rs5743836 allele A among patients.

Response: Agree with the respected reviewer. The sentence has been rephrased into “how a statistically significant higher frequency of rs5743836 allele T among patients “(l. 204-205 in revised version with track changes and l. 193-194 in clean corrected version).

Point 5: In fact, a qui-square testing with Yate’s correction resulted in a p = 0.031. Please, review calculations.

Response: agree with the respected reviewer, the following is the edited Table 2 with corrected numbers

Table 2. Allele and genotype frequency in cases and controls. 

SNP rs5743836 Case: N (%) Control: N (%) P-value

Allele A 213 (82) 412 (88) 0.031

Allele G 47 (18) 56 (12) 

Genotype A/A 87 (66.9) 183 (78.2) 0.062

Genotype A/G 39 (30) 46 (19.7) 

Genotype G/G 4 (3.1) 5 (2.1) 

Point 6: Considering the acknowledged relation of EBV infection and HL development, I would expect data concerning EBV positivity in this cohort.

Response: Unfortunately, the EBV status among our study patients’ group was not reported in patients’ medical records/pathology reports.

Point 7: As highlighted by the authors, sample size is extremely low. As comparison, the reported study from Carvalho et al, back in the 2012’s included three distinct cohorts encompassing respectively 797 Portuguese, 494 Italian and 868 US patients. In this context, besides significance, I wonder about the originality/novelty of the present study.

Response: We agree with the respected reviewer that the sample size is small. However; the respected reviewer comparison with the reported study from Carvalho et al is invalid as Carvalho et al study is dealing with non-Hodgkin lymphoma (NHL) cases and our study is dealing with Hodgkin lymphoma (HL) cases. By reviewing the reported studies of r5743836 polymorphism in TLR9 among HL patients, we found that the sample size of our study is among the largest. We think that the efficacy of the study is still statistically valid.

Response to Reviewer 2 Comments

We appreciate the respected reviewer comments. The following are our point-by- point responses.

Point: Please mention the possible relationship between genotypes and clinical parameters. And also you should give information about how this base change affects the TLR-9 gene function.

Response 1: Thank you for pointing this out. Regarding studying the possible relationship between genotypes and clinical parameters, unfortunately that was not statistically possible because all samples when sent to the Australian Genome Research Facility (AGRF, Melbourne Node, Melbourne, Australia) for genotyping of SNP rs5743836 (TLR9-1237T>C) were sent in anonymous order not linked to corresponding clinical data.

For information about how base change affects the TLR-9 gene function, the following sentence has been added “ People carrying the variant “C” allele of rs5743836 TLR9 promoter polymorphism (TLR9-123C) show increased TLR9 mRNA expression, transcriptional activity and deregulated immunological response secondary to increased binding affinity of TLR9 to NF-κB by creation of a potential NF-κB binding site (34)” (l. 115-118 in in revised version with track changes and l. 104-107 in clean corrected version).

---

## [Decision Letter · Decision Letter 1]

18 Jul 2022

Association of TLR9-1237T>C; rs5743836 polymorphism with increased risk of Hodgkin’s Lymphoma: a case-control study

PONE-D-22-13603R1

Dear Dr. Al Khatib,

We’re pleased to inform you that your manuscript has been judged scientifically suitable for publication and will be formally accepted for publication once it meets all outstanding technical requirements.

Kind regards,

Alvaro Galli

Academic Editor

PLOS ONE

Additional Editor Comments (optional):

Reviewers' comments:

Reviewer's Responses to Questions

**Comments to the Author**

1. If the authors have adequately addressed your comments raised in a previous round of review and you feel that this manuscript is now acceptable for publication, you may indicate that here to bypass the “Comments to the Author” section, enter your conflict of interest statement in the “Confidential to Editor” section, and submit your "Accept" recommendation.

Reviewer #1: All comments have been addressed

2. Is the manuscript technically sound, and do the data support the conclusions?

Reviewer #1: Yes

3. Has the statistical analysis been performed appropriately and rigorously? 

Reviewer #1: Yes

4. Have the authors made all data underlying the findings in their manuscript fully available?

Reviewer #1: Yes

5. Is the manuscript presented in an intelligible fashion and written in standard English?

Reviewer #1: Yes

6. Review Comments to the Author

Reviewer #1: All comments previously made by this reviewer were adequately approached in this new version. Specifically, the manuscript title was changed; a mention to studies who did not found association between the evaluated gene polymorphism and HD was added; the choice for this specific polymorphism was justified at Introduction with data from the literature; some misleading nomenclature were corrected; and some statistical data were also revised. Concerning the absence of data about EBV positivity in this cohort, I understand that authors are not able to come back to the original patients to include this point, and I also agree with authors concerning comparisons with the study from Carvalho et al.

Minor points that I would suggest, would be the inclusion of data from Table 1 at the text (under the section Material and Methods), and a review of the English-language usage and grammar, although these two points should not hinder the publication, since all methodological and scientific aspects are adequate.

7. PLOS authors have the option to publish the peer review history of their article (what does this mean?). If published, this will include your full peer review and any attached files.

Reviewer #1: No

---

## [Editor Report · Acceptance letter]

21 Jul 2022

PONE-D-22-13603R1 

Association of TLR9-1237T>C; rs5743836 polymorphism with increased risk of Hodgkin’s Lymphoma: a case-control study   

Dear Dr. Al-Khatib:

I'm pleased to inform you that your manuscript has been deemed suitable for publication in PLOS ONE. Congratulations! Your manuscript is now with our production department. 

Kind regards, 

on behalf of

Dr. Alvaro Galli 

Academic Editor

PLOS ONE